# Force Shadows: An Online Method to Estimate and Distribute Vertical Ground Reaction Forces from Kinematic Data

**DOI:** 10.3390/s20195709

**Published:** 2020-10-08

**Authors:** Alexander Weidmann, Bertram Taetz, Matthias Andres, Felix Laufer, Gabriele Bleser

**Affiliations:** 1Junior Research Group wearHEALTH, Technische Universität Kaiserslautern, Gottlieb-Daimler-Str. 48, 67663 Kaiserslautern, Germany; laufer@cs.uni-kl.de (F.L.); bleser@cs.uni-kl.de (G.B.); 2Augmented Vision Department, Deutsches Forschungszentrum für Künstliche Intelligenz (DFKI), Trippstadter Str. 122, 67663 Kaiserslautern, Germany; Bertram.Taetz@dfki.de; 3Department of Technomathematics, Technische Universität Kaiserslautern, Gottlieb-Daimler-Str. 48, 67663 Kaiserslautern, Germany; mandres@mathematik.uni-kl.de

**Keywords:** ground reaction force, vertical ground reaction force, vertical ground reaction force prediction, mobile kinetic analysis, inertial motion capture, pressure, plantar pressure distribution

## Abstract

Kinetic models of human motion rely on boundary conditions which are defined by the interaction of the body with its environment. In the simplest case, this interaction is limited to the foot contact with the ground and is given by the so called ground reaction force (GRF). A major challenge in the reconstruction of GRF from kinematic data is the double support phase, referring to the state with multiple ground contacts. In this case, the GRF prediction is not well defined. In this work we present an approach to reconstruct and distribute vertical GRF (vGRF) to each foot separately, using only kinematic data. We propose the biomechanically inspired force shadow method (FSM) to obtain a unique solution for any contact phase, including double support, of an arbitrary motion. We create a kinematic based function, model an anatomical foot shape and mimic the effect of hip muscle activations. We compare our estimations with the measurements of a Zebris pressure plate and obtain correlations of 0.39≤r≤0.94 for double support motions and 0.83≤r≤0.87 for a walking motion. The presented data is based on inertial human motion capture, showing the applicability for scenarios outside the laboratory. The proposed approach has low computational complexity and allows for online vGRF estimation.

## 1. Introduction

The analysis of forces driving the human motion and the forces exerted to the surrounding via contacts with the human body are part of kinetic human motion analysis, which is an important tool for instrumented human motion analysis [1,2]. According to Newton’s third law, the ground exerts a force on the human body by mirroring the acceleration of the body’s center of mass (CoM). This force is called *ground reaction force* (GRF) and carries important information to analyze gait parameters, e.g., in rehabilitation, sports, robotics, etc. [3]. The peak vertical GRF is a reliable parameter to evaluate lower extremity functional strength during sport-specific movements and can be used, together with the vertical impulse, which is the product of force multiplied by time during the propulsion phase of a vertical jump, for prognoses after sports-related injuries [4]. The landing phase in running sports produce high-frequency peaks in the vertical GRF trajectories and are assumed to play a crucial role in the development of injuries [5]. The vertical GRF can be used for online estimation of (unknown) carried loads, e.g., in working environments, in order to assess physical stress [6,7,8]. In laboratories, e.g., when human gait is measured and analyzed using force plates, subjects often need to strike a locally fixed measuring device during the measurement which can lead to disturbances in their natural walking behavior [9,10]. To avoid the restriction induced by a force plate, inverse dynamics can be used to derive kinetic parameters from kinematic data in order to estimate the current stress on joints during human motion and on an external contact point. Since kinematic data can be measured from mobile inertial motion capture, this allows for kinetic analysis outside the laboratory. The starting point for inverse dynamics calculations is the knowledge about the external forces. In this work, we focus on GRF, especially the vertical component. In the case of only one connected contact region, e.g., during single support phase in walking, the GRF can be uniquely derived from motion data, cf. [11,12,13]. However, in double support phases, more than one contact region is present and the human body forms a closed kinetic chain which makes the estimation of GRFs indeterminate (indeterminancy problem) [14,15,16]. In that case, we need to provide additional information to obtain a unique solution for the GRF. This was done via a smooth transition assumption and an empirically tuned function for gait in [11,13]. Static optimization and a detailed musculoskeletal model with additional muscle actuators under the feet were used in [14], which yielded good results for different types of motion; however, it has a very high computational cost, preventing online usage to date. A neural network was used in [10], which has low computational cost but is inherently data dependent. A further approach is to employ additional measurements for the vGRF via pressure insoles, cf. [17].

In contrast to the mentioned work, we propose a method that is independent of the motion and generalizes well to different motions, thus is not restricted to walking. Additionally, it has a low computational complexity which makes it real-time capable. The overall idea is to compute a conservative load distribution on each contact region (here, the feet) via a suitable projection of each segment along the negative GRF vector to the next surface (here, the ground) and utilize anatomical shape information of the foot as well as rough modeling of hip flexion as additional information. This allows for a conservative and consistent distribution of the vGRF in the double stance phase. The method is evaluated with three subjects that performed weight shifting movements with stationary placements of the feet (double support) as well as normal gait cycles on a single pressure plate.

The paper is structured as follows: Section 2 introduces the force shadow method (FSM), Section 3 describes the experimental results and Section 4 presents our conclusions as well as future work.

## 2. Methods

In this section, we formalize the underlying idea behind the FSM. According to Newton’s third law of motion, each body that exerts a force on a second one receives a force equal in magnitude and opposite direction. With this law in mind, and assuming that a standing person is in contact with the ground only via the feet, we observe an exerted force on the ground caused by the subject’s body weight, which in turn causes the ground to react with a force in opposite direction. This reaction force which acts on the human body (likewise in the dynamic case) is called the ground reaction force $F_{\mathrm{GRF}}\in {I\;R}^{3}$.

One key question motivating this work is the following: How can the GRF be consistently distributed over multiple contact regions, if only the kinematics and mass distributions of the body are present? The approach should be simple and consistent, i.e., match direct measurements at the contact regions.

One simple idea for the GRF estimation is to use only the center of mass (CoM) and compute the GRF with respect to one contact point [18]. This simplifies the kinematics of the complete human body to the kinematics of the CoM and reduces the interaction surface to one contact point, but does not allow to deal with multiple contact points.

The basic idea behind the FSM is to approximate a body mass distribution and project it onto a possible contact or interaction surface along the overall force vector, similar to a “shadow”. In a further step, this projected distribution is considered as weighting for the contact regions.

In Section 2.1, we formally explain how to determine $F_{\mathrm{GRF}}$ and how we use it to construct a force shadow (surface). As a direct application of the force shadow, Section 2.1.4 offers a general approach to obtain a plantar distribution of the total mass. In Section 2.2, we adjust the approach and make it more suitable to the human foot structure by incorporating the curvature of the foot arch and hip flexor muscles and propose the adjusted formula in Section 2.2.3. Then, we complete the method chapter presenting the algorithm in Section 2.4. The notation is explained in the abbreviations section, at the end of this article.

### 2.1. Force Shadow Method

Let us first start with formalizing the GRF with respect to the CoM of a multi-body system. In the following, we assume the positions x of the CoM and all segments to be given with respect to one global coordinate system *G*, i.e., $x=x^{G}$, but we leave out the superscript for better readability.

#### 2.1.1. Ground Reaction Force

Newton’s third law tells us that the ground reaction force mirrors every acceleration of the body’s CoM, based on the assumption that the feet define the only contact regions between the body and its surrounding. Let S be the set of all body segments, i.e., we assume that the body is a multi-body system consisting of $\left|S\right|$ rigid body segments with masses $m_{s}$, then the body’s CoM, $x_{\mathrm{CoM}}\in {I\;R}^{3}$, can be written as the weighted sum of the segments’ CoMs which can be derived as follows (cf. [19]):
(1)$$\begin{array}{cc} x_{\mathrm{CoM}} & =\frac{\int_{\mathrm{vol(body)}}x\;dm}{\int_{\mathrm{vol(body)}}\;dm}=\frac{\sum_{s\in S}m_{s}x_{{\mathrm{CoM}}_{s}}}{m_{\mathrm{total}}}, \end{array}$$
(2)$$\begin{array}{cc} m_{\mathrm{total}} & =\sum_{s\in S}m_{s}, \end{array}$$
where $m_{\mathrm{total}}$ denotes the subject’s total mass. Provided that we know the accelerations of each segment, e.g., by kinematics, Equation (2) allows to compute the acceleration of the body’s CoM:
$$\begin{array}{cc} {\ddot{x}}_{\mathrm{CoM}} & =\frac{\sum_{s\in S}m_{s}{\ddot{x}}_{{\mathrm{CoM}}_{s}}}{m_{\mathrm{total}}}. \end{array}$$


The ground reaction force can be derived as follows:
(3)$$\begin{array}{cc} F_{\mathrm{GRF}} & =-m_{\mathrm{total}}\left({\ddot{x}}_{\mathrm{CoM}}+{\left[\begin{array}{ccc} 0 & 0 & -g \end{array}\right]}^{T}\right). \end{array}$$


#### 2.1.2. Construction of Force Shadows for a Human Body

The force shadow method (FSM) constructs a surface by summing up multiple bivariate elliptical normal distributions using projected CoMs as mean value and the segments’ orientations to construct the corresponding covariances.

Let $x_{{\mathrm{CoM}}_{s},t}\in {I\;R}^{3}$ be the CoM for an arbitrary segment s∈S at time *t* which is assumed to be in the middle of the segment. The time stamp $t\in {I\;R}_{>0}$ denotes the current time at which the measurement was taken. Then, $x_{{\mathrm{CoM}}_{s},t}$ is projected onto the bottom (we assume z=0) along the ground reaction force vector in order to acquire the respective projected Gaussian center, denoted as $\mu_{s,t}\in {I\;R}^{3}$, i.e.,
(4)$$\begin{array}{cc} \mu_{s,t} & =x_{{\mathrm{CoM}}_{s},t}-kF_{\mathrm{GRF}}, \end{array}$$
(5)$$\begin{array}{cc} k & =\frac{x_{{\mathrm{CoM}}_{s},t}^{T}n}{F_{\mathrm{GRF}}^{T}n}, \end{array}$$
where $n={\left[0,0,1\right]}^{T}$ is the unit vector which is orthogonal to the ground. The factor k∈IR is chosen, such that the vertical component of $\mu_{s,t}$ is zero, i.e., it corresponds to a point on the interaction surface. This allows to consider variations in the 2-dimensional surface since only the projection onto the bottom plane is of interest, hence, we assume $x,\mu_{s,t}\in {I\;R}^{2}$ hereafter.

The covariance matrix $\Sigma_{s,t}\in {I\;R}^{2\times 2}$ is constructed in such a way that its principal components have the same orientation as the corresponding segment.

Let $a_{s},b_{s}\in I\;R$ be the parameters representing the dispersion of the ellipsoid of the Gaussian function and $R_{s,t}\in {I\;R}^{2\times 2}$ the rotation matrix responsible to rotate this ellipsoid around the vertical axis into the direction of the corresponding segment’s orientation, then the covariance is determined via
(6)$$\begin{array}{cc} \Sigma_{s,t} & :=\frac{C^{T}C}{\sqrt{{∥C∥}_{2}}}, \end{array}$$
(7)$$\begin{array}{cc} C & :=\left[\begin{array}{cc} a_{s} & 0\\ 0 & b_{s} \end{array}\right]R_{s,t}^{T}. \end{array}$$


See Section 2.3 for the estimation of $a_{s},b_{s}$. Equation (Equation 6) ensures the symmetry and the positive definiteness of the covariance matrix. Thus, for segment s∈S we obtain the function
(8)$$\begin{array}{cc} f_{s,t}\left(x\right): & =c_{s,t}N\left(x;\mu_{s,t},\Sigma_{s,t}\right),\;\mathrm{where}\; \end{array}$$
(9)$$\begin{array}{cc} N(x;\mu_{s,t},\Sigma_{s,t}) & =\frac{1}{2\pi \sqrt{\det\Sigma_{s,t}}}\exp\left(-\frac{1}{2}{\left(x-\mu_{s,t}\right)}^{T}\Sigma_{s,t}^{-1}\left(x-\mu_{s,t}\right)\right), \end{array}$$
(10)$$\begin{array}{cc} c_{s,t} & =\frac{d_{s}}{N(\mu_{s,t};\mu_{s,t},\Sigma_{s,t})}. \end{array}$$


The constant $d_{s}\in \left[0,1\right]$ represents the contribution of segment *s* to the total body mass in terms of body mass proportion as proposed in [20], they are given in percentage. We repeat the projection for all segments. The *shadow function* is then obtained via superposition of all projected contributions of the segments
(11)$$\begin{array}{cc} f_{t}\left(x\right) & :=\sum_{s\in S}f_{s,t}\left(x\right). \end{array}$$


For an illustration, see Figure 1.

#### 2.1.3. Contact Region Estimation

In order to obtain contact regions for a general motion, we need a model for the respective parts of the body that incorporates possible contact with the environment. In this work, we consider only the foot regions, depicted in Figure 1, and a flat ground as interaction surface. The foot region model is registered with a 2D foot model that is tailored to the person at hand.

The pose of the region model is computed as follows: Note, in the following, we consider the left foot only; the right foot can be treated analogously. We denote $x_{i}\in {I\;R}^{2}$ for $i\in LM:=\left\{FM,VM,CA\right\}$ as the anatomic landmarks taken from the current pose and $p_{i}\in {I\;R}^{2}$ the respective landmarks of the foot model; then we minimize the energy function
(12)$$\begin{array}{c} E\left(M,t\right):=\sum_{i\in LM}{∥\left(Mp_{i}+t\right)-x_{i}∥}_{2}, \end{array}$$
where $t\in {I\;R}^{2}$, $M\in {I\;R}^{2\times 2}$, in order to obtain a transformation of the foot model, see Figure 2.

We consider the transformation as a matrix multiplication using homogeneous coordinates, i.e.,
(13)$$\begin{array}{c} T:=\left[\begin{array}{cc} M & t\\ 0^{T} & 1 \end{array}\right]\in {I\;R}^{3\times 3}, \end{array}$$
where $0={\left[0,0\right]}^{T}$ is a column vector consisting of zeros. In order to apply the transformation to the points, these are transformed into homogeneous coordinates by adding 1 to the third component, i.e., ${\bar{x}}_{i}:={\left[\begin{array}{cc} x_{i}^{T} & 1 \end{array}\right]}^{T}\in {I\;R}^{3}$, ${\bar{p}}_{i}:={\left[\begin{array}{cc} p_{i}^{T} & 1 \end{array}\right]}^{T}\in {I\;R}^{3}$, for all i∈LM.

Let $X:=\left[\begin{array}{ccc} {\bar{x}}_{FM} & {\bar{x}}_{VM} & {\bar{x}}_{CA} \end{array}\right]$ and $P:=\left[\begin{array}{ccc} {\bar{p}}_{FM} & {\bar{p}}_{VM} & {\bar{p}}_{CA} \end{array}\right]$, $X,P\in {I\;R}^{3\times 3}$, be the homogeneous marker points in matrix form. Then, we can rewrite Equation (Equation 12) as
(14)$$\begin{array}{c} E\left(T\right)={∥TP-X∥}_{2}. \end{array}$$


The linear least-squares problem
(15)$$\begin{array}{c} \min_{T}E\left(T\right), \end{array}$$
can be solved in closed-form via the pseudo-inverse, e.g., utilizing the singular value decomposition (SVD). In each time step, the regions $\Omega_{L},\Omega_{R}$ are transformed this way.

The (sub-)regions in $\Omega_{L},\Omega_{R}$ are defined to be in contact with the ground if the registered center falls below a distance $h_{\min}\approx 0.03\;m$ from the ground and has a velocity below $v_{min}\approx 0.8\;\frac{m}{s}$. These values are chosen similar to [14]. If, for a (sub-)region, at least one condition (height below $h_{min}$ or velocity below $v_{min}$) is not fulfilled, this particular region is assumed to have no ground contact and, therefore, set to zero. With these conditions, the method is independent of stance times or gait event patterns.

#### 2.1.4. Force Distribution

Assuming unconnected sets of regions $\Omega_{L}\subseteq {I\;R}^{2}$ for the left and $\Omega_{R}\subseteq {I\;R}^{2}$ for the right foot obtained from the registration described in the previous section: Consider an arbitrary region $\Omega_{k}\subseteq \Omega_{L}$ of the left foot model ($\Omega_{k}\subseteq \Omega_{R}$ analogously). Let $F_{\mathrm{GRF},z}$ be the vertical component of the ground reaction force from Equation (Equation 3), then, the estimated load on this region is generally given by
(16)$$\begin{array}{cc} L_{\Omega_{k},t} & =F_{\mathrm{GRF},z}I_{\Omega_{k},t}, \end{array}$$
where the load distribution factor $I_{\Omega_{k},t}\in \left[0,1\right]$ for the corresponding region is computed by integration and normalization, i.e,
(17)$$\begin{array}{cc} I_{\Omega_{k},t} & =\frac{\int_{\Omega_{k}}f_{t}\left(x\right)dx}{\int_{\Omega_{L}}f_{t}\left(x\right)dx+\int_{\Omega_{R}}f_{t}\left(x\right)dx}. \end{array}$$


Here, $f_{t}$ corresponds to the force shadow defined in (Equation 11). The integral of an arbitrary region $\Omega_{k}\subseteq \left(\Omega_{L}\cup \Omega_{R}\right)$ is approximated via a triangulation of this region (see Figure 1 on the right) using the midpoint rule, that is,
(18)$$\begin{array}{c} \int_{\Omega_{k}}f_{t}\left(x\right)dx\approx \sum_{i=1}^{N_{k}^{\Delta }}\overset{{\bar{f}}_{i,t}}{\overset{︷}{\left(\frac{1}{3}\sum_{j\in \mathrm{Nodes}(\Delta_{i})}f_{t}(x_{j})\right)}}\left|\Delta_{i}\right|, \end{array}$$
where ${\bar{f}}_{i,t}$ is the mean value of the points evaluated at the nodes and $\left|\Delta_{i}\right|$ is the area of the triangle $\Delta_{i}$. The number of triangles in region $\Omega_{k}$ is denoted as $N_{k}^{\Delta }$.

### 2.2. Anatomical Models

In this section, we describe additional, biomechanically motivated modifications of the load distribution.

#### 2.2.1. Arch of Foot

The above described force distribution method puts equal weight on all points of the foot region model. This leads to a similar load, e.g., on the heel and the arch, during a neutral stance which would be only the case for a pathologic foot anatomy. This would have an impact on the center of pressure (CoP) for each foot, i.e., on the vGRFs’ application points. To compensate for this effect, we take additional weights into account. To model the curvature of the foot, we fit a surface
(19)$$\begin{array}{cc} w(x) & =a_{0}+a_{1}x+a_{2}y+a_{3}xy+a_{4}x^{2}+a_{5}y^{2}, \end{array}$$
for $x:=\left[x,y\right]\in {I\;R}^{2}$, such that w(x)=1 for the contact points of the metatarsal heads and the calcaneus bone and $w\left(x\right)=h_{a}dist_{\mathrm{CP}}$ for the midpoints between the front and rear contact points, where $dist_{\mathrm{CP}}:={∥x_{CA}-x_{VM}∥}_{2}$ is the distance between the the subject’s fifth metatarsal head and calcaneus (not to be confused with the fixed scalar $d_{CP}$ from Figure 3). The scalar factor $h_{a}\in I\;R$ is a parameter modeling the anatomical height of the cuboid bone relative to the distance between the support points of calcaneus and the head of the fifth metatarsal bone, cf. Figure 3 on the left.

To guarantee the conservation of integrals, we rescale the weight function pointwise with a time-dependent scalar
(20)$$\begin{array}{cc} {\tilde{w}}_{t}\left(x\right) & :=n_{t}\;w\left(x\right)\\ n_{t} & =\frac{\int_{\Omega }f_{t}\left(x\right)dx}{\int_{\Omega }w\left(x\right)f_{t}\left(x\right)dx}\in I\;R. \end{array}$$


Note that the scalars $n_{t}$ belonging to the left and right foot differ from each other, thus, we have to do this twice to obtain ${\tilde{w}}_{L,t}\left(x\right)$ and ${\tilde{w}}_{R,t}\left(x\right)$.

#### 2.2.2. Hip Flexion Muscles

So far, the proposed approach cannot detect decreasing loads resulting from lifting off the feet slightly, for example, in double support movements in which the whole CoM is located within the region of one foot. This can hardly be measured by human motion capturing systems, especially when the feet do not lift off, but influences the vGRF. Therefore, we model the effect of the muscles responsible for hip flexion, see Figure 3 on the right, as follows: We assume that the muscles of the unloaded leg are increasingly activated, the longer the mass is balanced on the other foot, which is interpretable as raising the foot gradually, the longer the person stands on the other foot.

In order to model the described effect, the total loads are distributed accordingly to the left and right foot. Therefore, for each time step $t\in {I\;R}_{>0}$ we compute the current velocity, ${\dot{x}}_{\mathrm{CoM},t}$, of the CoM to predict its position for the next $\bar{t}\in I\;N$ time instances, i.e.,
(21)$$\begin{array}{cc} {\tilde{x}}_{\mathrm{CoM},t+k} & :=x_{\mathrm{CoM},t}+k\Delta t\cdot {\dot{x}}_{\mathrm{CoM},t},\;\mathrm{for}\;k\in \left\{0,\ldots ,\bar{t}\right\}\subseteq I\;N. \end{array}$$


Let $s_{L,t},s_{R,t}\in \left[0,1\right]$, initialized as $s_{L,t_{0}},s_{R,t_{0}}=1$, be the distribution parameters for the left and right foot. Then, if one of the predicted CoMs is contained in the convex hull of the left foot, that is, ${\tilde{x}}_{\mathrm{CoM},t+k}\in Conv\left(\Omega_{L}\right)$, for at least one *k*, we increase $s_{L,t}$ (if $s_{L,t}<1$) and decrease $s_{R,t}$ (if $s_{R,t}>0$) using a ramp function with slope *m* (see Section 2.3 for the estimation of *m*). There are three cases in total which can occur:
$$\begin{array}{cc} s_{L,t}↗,s_{R,t}↘ & \;\;\;\;\;\;\mathrm{if}\;\mathrm{there}\;\mathrm{is}\;a\;k,\;\mathrm{such}\;\mathrm{that}\;{\tilde{x}}_{\mathrm{CoM},t+k}\in Conv\left(\Omega_{L}\right),\\ s_{L,t}↗,s_{R,t}↗ & \;\;\;\;\;\;\mathrm{if}\;\mathrm{for}\;\mathrm{all}\;k:{\tilde{x}}_{\mathrm{CoM},t+k}\notin Conv\left(\Omega_{L}\right)\\ \; & \;\;\;\;\;\;\mathrm{and}\;\;\;\;\;\;{\tilde{x}}_{\mathrm{CoM},t+k}\notin Conv\left(\Omega_{R}\right),\\ s_{L,t}↘,s_{R,t}↗ & \;\;\;\;\;\;\mathrm{if}\;\mathrm{there}\;\mathrm{is}\;a\;k,\;\mathrm{such}\;\mathrm{that}\;{\tilde{x}}_{\mathrm{CoM},t+k}\in Conv\left(\Omega_{R}\right). \end{array}$$


Note that the increment and decrement should provide that $s_{L,t},s_{R,t}\in \left[0,1\right]$, e.g., we do not increase $s_{L,t_{0}},s_{R,t_{0}}$ as they already are at their maximum. The weights ${\tilde{w}}_{t}\left(x\right)$ from (Equation 20) are then multiplied by a linear combination of $s_{L,t}$ and $s_{R,t}$, which leads to the modified weight functions
(22)$$\begin{array}{l} {\hat{w}}_{L,t}\left(x\right)={\tilde{w}}_{L,t}\left(x\right)\left(s_{L,t}+\left(1-s_{R,t}\right)\right),\\ {\hat{w}}_{R,t}\left(x\right)={\tilde{w}}_{R,t}\left(x\right)\left(s_{R,t}+\left(1-s_{L,t}\right)\right). \end{array}$$


#### 2.2.3. Anatomy Aware Load Estimation

Incorporating the anatomy of a normal foot and mimicking the effect of hip flexion muscles on the FSM in (Equation 16) and (Equation 17) for an arbitrary region $\Omega_{k}$ we obtain the final vGRF distribution formula as follows
(23)$$\begin{array}{cc} L_{\Omega_{k},t} & =F_{\mathrm{GRF},z}\frac{\int_{\Omega_{k}}{\hat{w}}_{L,t}\left(x\right)f_{t}\left(x\right)dx}{\int_{\Omega_{L}}{\hat{w}}_{L,t}\left(x\right)f_{t}\left(x\right)dx+\int_{\Omega_{R}}{\hat{w}}_{R,t}\left(x\right)f_{t}\left(x\right)dx}. \end{array}$$


### 2.3. Utilized Hyper Parameters

For optimizing the hyper parameters introduced in the previous sections, i.e., the dispersion parameters $a_{s},b_{s}$ constructing the Gaussian bell functions in Equation (Equation 7), the time parameter $\bar{t}$, and the slope *m* of the ramp in Section 2.2.2, we used the load trajectories of a representative dataset (we choose sway all around (AR) of one representative person, see Section 3.2 for the description of the performed motions) for each of the six depicted areas of the foot model, see Figure 1 on the right; that is, we define the vector $L_{\Omega_{L},i_{t}},L_{\Omega_{R},i_{t}}\in {I\;R}^{6}$ as the estimated load vector for the time frame $i_{t}$ and $P_{\Omega_{L},i_{t}},P_{\Omega_{R},i_{t}}\in {I\;R}^{6}$ regarding the corresponding loads measured by the pressure plate. Let $N_{f}$ denote the number of frames of the trajectories and P the set of all parameters, i.e.,
(24)$$\begin{array}{c} P:=\left\{(a_{s},b_{s})|s\in S\right\}\cup \left\{\bar{t},m\right\}, \end{array}$$
then, we minimize the following energy function
$$\begin{array}{cc} E(P) & :=\sqrt{\sum_{i_{t}=1}^{N_{f}}\left({∥L_{\Omega_{L},i_{t}}\left(P\right)\;-P_{\Omega_{L},i_{t}}\;∥}_{2}^{2}+{∥L_{\Omega_{R},i_{t}}\left(P\right)\;-P_{\Omega_{R},i_{t}}\;∥}_{2}^{2}\right)} \end{array}$$
with respect to P using the interior-point method with a step tolerance of $10^{-8}$, that is, the optimization ends if $∥P_{k+1}-P_{k}∥<10^{-8}\left(1+∥P_{k}∥\right)$. Note the dependency of the FSM on the parameter set, i.e., $L_{\Omega_{L},i_{t}}=L_{\Omega_{L},i_{t}}\left(P\right)$ and $L_{\Omega_{R},i_{t}}=L_{\Omega_{R},i_{t}}\left(P\right)$.

### 2.4. FSM Algorithm

The subsequent algorithm summarizes the proposed method and requirements. We denote the locations of the foot landmarks as $x_{FM},x_{VM},x_{CA}$ and $p_{FM},p_{VM},p_{CA}$ regarding the kinematic pose and the foot model, respectively (cf. Figure 1).

## 3. Experimental Results and Discussion

### 3.1. Materials

For the validation of the proposed method, we used two measurement systems. The Xsens full body inertial measurement system with 17 IMUs at a sampling rate of 60 Hz to capture human motion; MVN studio 4.97.1 for data acquisition. The validated Zebris pressure plate type FDM 1.5 [21] acted as reference measuring system for pressure values. The pressure plate has an update rate of 100 Hz, a sensor area with dimensions 149 cm × 54.2 cm (length × width) and a spatial pressure value resolution of 0.8469 cm^2^, i.e., per time frame, the pressure plate data is stored in a 176 × 64 pressure matrix. The methods were implemented and evaluated in MATLAB, version 9.4.0.813654 (R2018a).

### 3.2. Experiments

Three subjects (body weight 80.3 ± 7.7 kg, i.e. between 73 and 87 kg, age between 27 and 29 years) walked onto the pressure plate and performed double support movements (weight shifting), i.e., sway all around (AR), sway from side to side (S2S) and sway back and forth (BF), with eighteen cycles (three cycles per measurement) in total for each movement. Further, more dynamic motions like squatting (SQ) and walking (NW) were performed. As in the double support case, eighteen squats (three squats per measurement) in total were measured. Regarding NW, the subjects walked over the plate in normal speed measuring nine non-consecutive gait cycles (heel-strike to toe-off) per measurement; in total, 54 gait cycles were measured. Due to poor inertial tracking data, 12 cycles were omitted.

After receiving all relevant study information, the participants signed an informed consent to the study including a permission to publish the data.

The focus is on the evaluation of the predicted vGRFs and the forces for the three regions (the front part is summarized into one) per foot shown in Figure 1 using the proposed method in Algorithm 1 as compared to the pressure plate data.

**Algorithm 1** Force Shadow Method (FSM)**Require:**$m_{\mathrm{total}}$, w(x) (cf. (Equation 19)), set of hyper parameters P;
  1:**Input:** kinematic data: $x_{{\mathrm{CoM}}_{s},t}$ for all segments s∈S, $x_{\mathrm{CoM},t}$, $x_{FM}$, $x_{VM}$, $x_{CA}$;  2:    foot model: triangle mesh grid, $p_{FM}$, $p_{VM}$, $p_{CA}$;  3:**Output:**$L_{\Omega_{k},t}\;\mathrm{for}\;\mathrm{all}\;\mathrm{regions}\;\Omega_{k}\subset \left(\Omega_{L}\cup \Omega_{R}\right)$;  4:**procedure**Force Shadow Construction  5:    Determine $F_{\mathrm{GRF}}$, cf. (Equation 3);  6:    Project $x_{{\mathrm{CoM}}_{s},t}$ along $F_{\mathrm{GRF}}$ to obtain $\mu_{s,t}$ and construct $\Sigma_{s,t}$ according to (Equation 4)–(Equation 7)  7:    Determine the shadow function $f_{t}\left(x\right)$ using $\mu_{s,t},\Sigma_{s,t}$, cf. (Equation 8)–(Equation 11);  8:**procedure**Weights incorporating anatomy  9:    Register foot models into the current kinematic pose using the foot landmarks, cf. (Equation 12)–(Equation 15); 10:    Determine ${\tilde{w}}_{t}\left(x\right)=n_{t}w\left(x\right)$, cf. (Equation 20); 11:    Project the current body’s center of mass $x_{\mathrm{CoM},t}$ along $F_{\mathrm{GRF}}$ onto the ground; 12:    Predict $x_{\mathrm{CoM},\tau }$ for $\tau \in [t,t+\bar{t}]$ to obtain modified weights ${\hat{w}}_{\cdot ,t}$, cf. (Equation 21) and (Equation 22); 13:**procedure**Integration 14:    Integrate the weighted force shadow function ${\hat{w}}_{\cdot ,t}f_{t}$ over the triangle meshes $\Omega_{k}$, cf. (Equation 18), (Equation 23); 15:**return:**$L_{\Omega_{k},t}\;\mathrm{for}\;\mathrm{all}\;\mathrm{regions}\;\Omega_{k}\subset \left(\Omega_{L}\cup \Omega_{R}\right)$;

### 3.3. Registration of the Pressure Plate Data

In order to obtain reference data from the pressure plate, the locations of the foot prints on the pressure plate need to be known, such that we can detect which region $\Omega_{k}$ is activated and which force is acting on this respective region. For that reason, we extract the foot prints on the pressure plate by summing up the 176 × 64 pressure matrices for every time frame (given a static pose of the feet, i.e., we know that in the measured range the subject’s foot pose was fixed) and use an indicator function for each activated cell, i.e., we set the cells with non-vanishing pressure values to one; that way, we obtain a binary matrix storing the spatial information about the 2D foot prints. The connected non-zero entries of the matrix represent the Cartesian coordinates of the foot prints. These coordinates are registered to the foot models via a 2D point registration, using the rigid extended coherent point drift (rECPD), cf. [22]. To determine the vGRF for the region $\Omega_{k}$ the pressure values that are active at the respective region and time are multiplied pointwise by the cell’s area (0.8469 cm^2^) and summed up in that region. For the total vGRF the total foot prints need to be considered.

### 3.4. Error Metrics

To evaluate the performance of the FSM, the estimations y(k),fork=1,…,N, were compared with the measurements $\tilde{y}\left(k\right)$ of the pressure plate, both normalized by bodyweight, e.g., $y_{\mathrm{Heel}}\left(k\right)=\frac{L_{\Omega_{\mathrm{Heel}},k}}{m_{\mathrm{total}}}$ regarding the estimation for the heel at time step *k*.

In order to enable the comparison to recent studies, we include the metrics defined in [13], i.e., the root-mean-square error (RMSE),
(25)$$\begin{array}{cc} \mathrm{RMSE} & =\sqrt{\frac{1}{N}\sum_{k=1}^{N}{\left(y\left(k\right)-\tilde{y}\left(k\right)\right)}^{2}}=\sqrt{\frac{1}{N}\sum_{k=1}^{N}e{\left(k\right)}^{2}}, \end{array}$$
where $e\left(k\right)=y\left(k\right)-\tilde{y}\left(k\right)$ is defined as the error at the $k^{th}$ sample, and the relative root-mean-square error (rRMSE),
(26)$$\begin{array}{cc} \mathrm{rRMSE} & =\frac{\mathrm{RMSE}}{\frac{1}{2}\left(\max_{k}\left(y\left(k\right)\right)-\min_{k}\left(y\left(k\right)\right)+\max_{k}\left(\tilde{y}\left(k\right)\right)-\min_{k}\left(\tilde{y}\left(k\right)\right)\right)}\times 100\%, \end{array}$$
which constitutes the normalization of RMSE by the peak-to-peak amplitude between both discrete trajectories.

Further metrics included in the evaluation are the mean absolute error (MAE) of the sample data
(27)$$\begin{array}{c} \mathrm{MAE}=\frac{1}{N}\sum_{k=1}^{N}\left|e(k)\right|, \end{array}$$
the standard deviation of the absolute error (SD)
(28)$$\begin{array}{c} \mathrm{SD}=\sqrt{\frac{1}{N-1}\sum_{k=1}^{N}{\left(\left|e(k)\right|-\bar{e}\right)}^{2}}, \end{array}$$
where
(29)$$\begin{array}{c} \bar{e}=\frac{1}{N}\sum_{k=1}^{N}e(k), \end{array}$$
and the Pearson correlation coefficient (*r*)
(30)$$\begin{array}{cc} r & =\frac{\sum_{k=1}^{N}\left(y\left(k\right)-\bar{y}\right)\left(\tilde{y}\left(k\right)-\bar{\tilde{y}}\right)}{\sqrt{\sum_{k=1}^{N}{\left(y\left(k\right)-\bar{y}\right)}^{2}}\sqrt{\sum_{k=1}^{N}{\left(\tilde{y}\left(k\right)-\bar{\tilde{y}}\right)}^{2}}}, \end{array}$$
where
(31)$$\begin{array}{cc} \bar{y} & =\frac{1}{N}\sum_{k=1}^{N}y(k),\;\bar{\tilde{y}}=\frac{1}{N}\sum_{k=1}^{N}\tilde{y}\left(k\right). \end{array}$$


### 3.5. Subregions

The results of the subregions in $\Omega_{L}$ and $\Omega_{R}$, according to the foot model shown in Figure 1 on the right, are listed in Table 1. The subregions comprise heel, arch of the foot and front foot, which consists of $\Omega_{\mathrm{Metatarsum}}$ and $\Omega_{\mathrm{Toes}}$.

The smallest deviations can be found in the mid foot with a RMSE of at most $0.76\;\frac{N}{\mathrm{kg}}$ for weight shifting movements and $1.03\;\frac{N}{\mathrm{kg}}$ for walking. The RMSEs of estimated loads on heel and front foot do not exceed $2.36\;\frac{N}{\mathrm{kg}}$ and $2.13\;\frac{N}{\mathrm{kg}}$, respectively, which can be found in S2S and NW movements. Regarding the entire foot area, the RMSE lies in between $0.60\;\frac{N}{\mathrm{kg}}$ and $1.64\;\frac{N}{\mathrm{kg}}$ for all movements. Small deviations in the CoM can already falsely detect the activation of hip flexion muscles leading to lower correlations, see the correlation coefficients in SQ. Hence, overall, the results show that the FSM can approximate measured pressure values and vGRF reasonably well, based on the kinematics of the human body, its CoM and the force shadow function.

### 3.6. Justification of Arch and Hip Flexion Model

Figure 4, Figure 5 and Figure 6 justify the importance of choosing elliptic Gaussian functions, modeling the arch of the foot and modeling the hip flexor muscles. The plots show the trajectories of the vGRF or ${\mathrm{vGRF}}_{Arch}$ for the left and right foot compared to the reference from the Zebris plate during double support movements with and without these extensions.

### 3.7. Gait Cycle Trajectories

Figure 7 shows the vGRF trajectories obtained from the FSM as compared to the reference data during walking with normal speed (normalized with the corresponding total weight). Shown are the gait cycles from heel-strike to toe-off. According to [13,23,24], the trajectories follow the typical structural pattern of gait cycles.

### 3.8. Execution Time

In Table 2, the execution times for the substeps of the proposed FSM algorithm are listed; some substeps are taken together due to interdependencies. As the computation time is independent of the movement, one AR measurement with 1308 time frames was chosen for the statistics. Note, the execution time was produced with (non-optimized) MATLAB code, which can be much slower as compared to an optimized C++ implementation. A single iteration of the algorithm takes 114.1 ms (average over 1308 frames) making the method usable for online applications. The algorithm was executed using an Intel Core i7-7700 processor at 3.6 GHz.

### 3.9. Discussion on Application Scenarios and Accuracy

The FSM predicts the vGRF for arbitrary contact regions online and is designed to be robust to different stance times or gait events, e.g., initial contacts; via biomechanically motivated models and few hyper-parameters. However, the method has not been validated for patients with severe gait abnormalities, so far. Possible differences to the presented results could arise, e.g., for different single stance times, due to abnormalities of gait or balance that could have an influence on the hip flexion model. The parameters $\bar{t}$ and *m* for the hip flexion model were treated as hyper-parameters in this work and were only tested for the presented motion types for healthy participants.

Regarding accuracy comparisons with existing methods for GRF prediction, the presented approach has comparable to inferior accuracy regarding vGRF prediction as compared to the approach presented in [14], with respect to the squatting motion and an increased error (by about a factor of 2) with respect to normal walking (cf. Table 3 of [14] with Table 1). Note that the approach [14] is an offline approach that considers an optimization over a detailed musculoskeletal model and is, thus, considerably more expensive regarding computation time and only applicable with the detailed musculoskeletal model. Moreover, our approach was not yet optimized for walking scenarios, e.g., the hyper-parameters were only tuned on one person for the AR motion and used for all other motions, including NW.

One application scenario, where the FSM can be applied, is related to the Ovako Working Posture Analysing System (OWAS) [6,7,8] to assess work postures. This system is partitioned in four parts, the first three parts are concerned with the worker’s pose, in particular, the pose of the head, back, arms and legs. The fourth OWAS category classifies the carried load in the following classes, from 0kg to 10kg, 10kg to 20kg and above 20kg with the respective scores. With this specific application in mind and assuming a target accuracy of 5kg to 10kg, we tolerate errors of about $0.61\;\frac{N}{\mathrm{kg}}$ to $1.22\;\frac{N}{\mathrm{kg}}$ (using the subjects’ average body weight of 80.3kg for weight normalization). Comparing with the results in Table 1, this target accuracy can be reached. However, to estimate unknown carried loads, additional data is needed, which can be provided by pressure insoles. In this scenario, the proposed method can act as a predictor for vGRF (in comparison to the the insoles’ measurements), especially when ambiguities due to increased accelerations (caused by motion) occur, as this would lead to ambiguities with increased loads, if measured with insoles alone.

### 3.10. Main Limitations

We used full body kinematics, based on inertial motion tracking, and a mass distribution table [20] to obtain the CoM kinematics. Inertial motion capture can have sensor-to-segment calibration errors with severe impacts on the tracked kinematics [25]. Figure 8 demonstrates how errors in tracking the CoM can result in vGRF deviations. In this situation, the FSM is not able to distribute the vGRF accordingly, especially in movements where the CoM is predominantly located in the center of the body, see the correlation coefficients for BF or SQ in Table 1.

Further, the pressure estimates of the different regions can be ambiguous, in the case where the pressure distribution under the foot is changed due to muscle contraction, but the kinematics do not change. Thus, the distributions under each foot have to be interpreted with care. However, the total vGRF, predicted for each independent contact region, e.g., left or right foot, is invariant to this due to integration over the region. The only variation here can happen in the application point of the vGRF, i.e., the CoP under the foot, which is derived from the distribution of the respective contact region.

Finally, the results have been, up to this point, only tested on healthy male subjects of rather similar weight.

## 4. Conclusions and Future Work

### 4.1. Conclusions

In this work, we presented the FSM which is an approach to estimate and distribute vGRF for multiple contact regions, using only kinematic data of a human body, approximate segment weight distributions and the person’s overall weight. This has been addressed in literature so far only for specific motion patterns [11,13], completely data-based, i.e., with neural networks [10], or with a high computational burden, utilizing a detailed musculoskeletal model [14,26]. In contrast, the proposed approach is generally applicable, model-based and has a small computational complexity which allows for online (real-time) usage. The experimental results also show that the proposed FSM can approximate measured pressure values and vGRF reasonably well with kinematic data from inertial body motion capture, hence, allowing for a mobile setup. Therefore, this work is meant as another step in the direction of consistent (online) mobile kinematic and kinetic analysis of human motion which is currently lacking in literature.

### 4.2. Future Work

The FSM is not limited to the shown movements and should be evaluated for other, more dynamic, movements, e.g., running and cutting maneuvers, which are relevant in the field of sports medicine.

In the current work, only healthy foot anatomy was considered. However, the anatomical model can be extended to pathologic feet.

The projected distribution parameters ($a_{s},b_{s}$) introduced in (7) are currently optimized to fit a small set of subjects. According to the constructive nature of the approach, these could be replaced by projected segment volumes of an adapted human body model, e.g., automatically generated based on a portable RGB-D sensor as proposed in [27,28,29].

The influence of sensor-to-segment calibration errors already described in Section 3.10 could be addressed by including automatic sensor alignment in the inertial tracking in future studies, similar to [30,31], or by applying other corrections of the body’s CoM. One possibility is the exploitation of sensor insoles to provide additional data. In this scenario, the FSM could be used as a predictor (in the sense of statistical inference) to improve the insoles’ measurements via bias estimation or estimate additionally carried loads, see also Section 3.9. The combination of full-body kinematics and carried load estimation is particularly interesting in the context of ergonomic assessment, see e.g., [6,7,8,32,33].

In order to get access to complete gait analyzes, it is inevitable to estimate the full 3D GRF and the ground reaction torques for each foot. Based on the latter, one can also obtain the center of pressure, which can be used as additional information to correct the CoM.

## Figures and Tables

**Figure 1 sensors-20-05709-f001:**
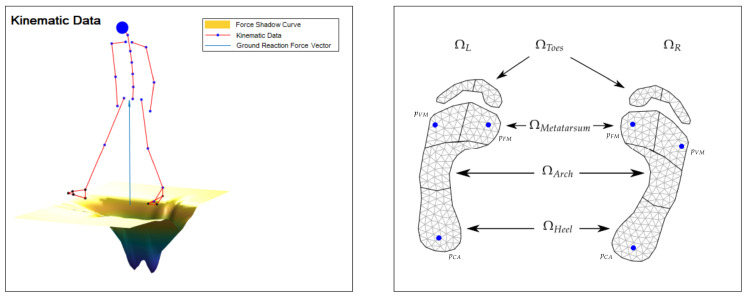
Kinematic data with constructed force shadow function underneath the feet; the blue vector describes the ground reaction force $F_{\mathrm{GRF}}$ (**left plot**). Standard foot model for the left and right foot with partitioned regions $\Omega_{i}$ (**right plot**); the location of the first- ($p_{FM}$) and the fifth metatarsal head ($p_{VM}$) and the calcaneus bone $\left(p_{CA}\right)$ are marked as blue points.

**Figure 2 sensors-20-05709-f002:**
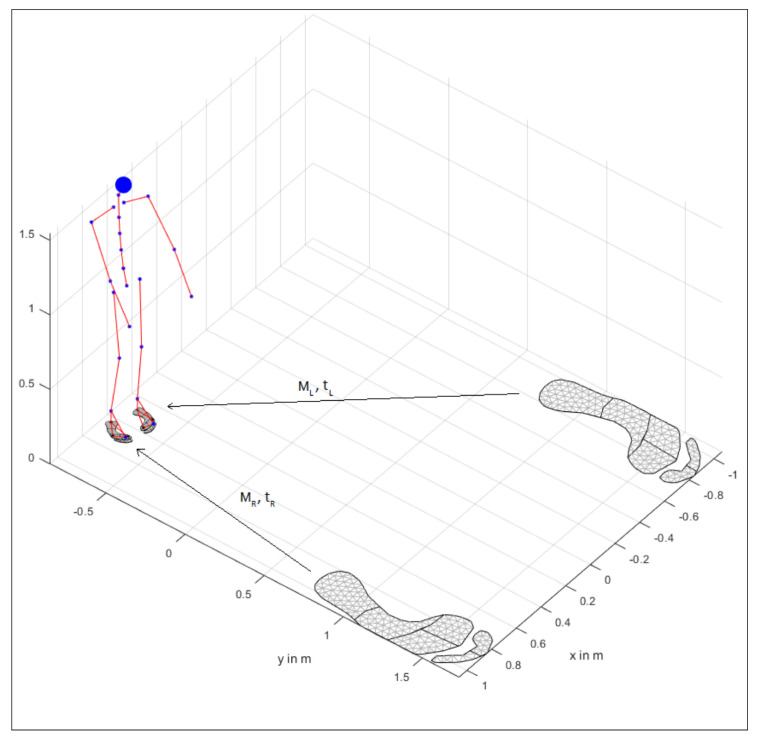
Standard foot model registered via right ($M_{R},t_{R}$) and left ($M_{L},t_{L}$) transformation using the anatomic landmarks from both the model and the kinematics.

**Figure 3 sensors-20-05709-f003:**
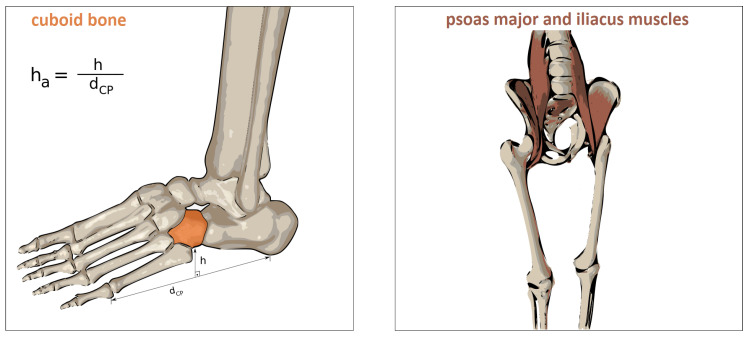
Left cuboid bone to model the arch of the foot using the relative height $h_{a}$ (**left**) and muscles responsible for hip flexion, i.e., psoas major and iliacus muscles (**right**).

**Figure 4 sensors-20-05709-f004:**
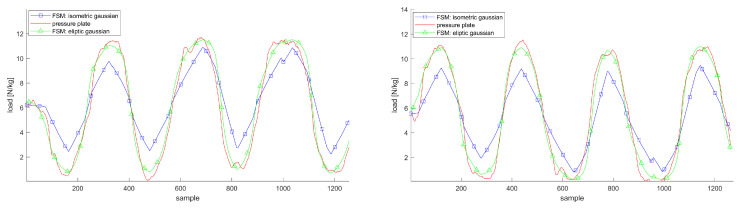
vGRF trajectories for the left and right foot applying isometric Gaussian functions, i.e., $a_{s},b_{s}=1,\;\mathrm{for}\;\mathrm{all}\;s\in S$, (blue squares) and optimized, elliptic functions (green triangles) during AR movement compared to the pressure plate (red line); both plots show the estimations and measurements in Newton per kilogram.

**Figure 5 sensors-20-05709-f005:**
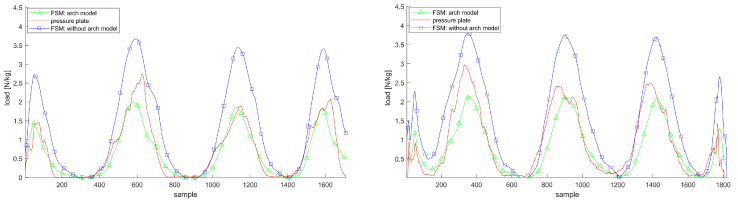
${\mathrm{vGRF}}_{\mathrm{Arch}}$ trajectories for the left and right foot before (blue squares) and after (green triangles) modeling the foot arch during AR movement (left) compared to the pressure plate (red line); both plots show the estimations and measurements in Newton per kilogram.

**Figure 6 sensors-20-05709-f006:**
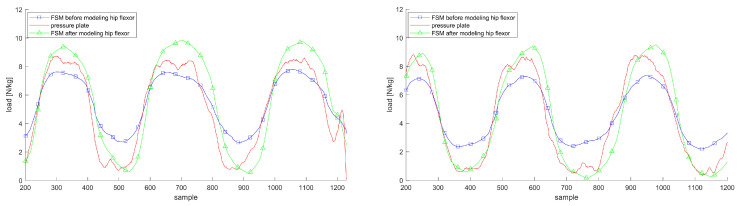
vGRF trajectories for the left and right foot before (blue squares) and after (green triangles) modeling the hip flexion compared to the pressure plate (red line) during AR movement; both plots show the estimations and measurements in Newton per kilogram.

**Figure 7 sensors-20-05709-f007:**
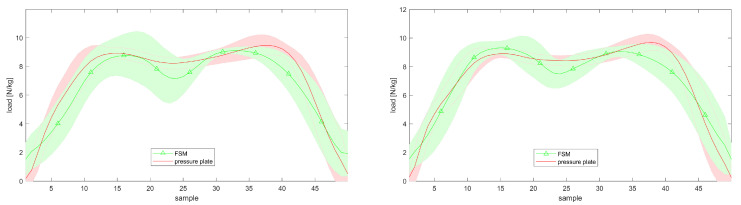
vGRF trajectories for the left and right foot resulting from FSM (green indication) compared to the pressure plate (red indication) normalized by the total body weight; the lines illustrate the mean vGRF values taken from multiple gait cycles (from heel-strike to toe-off) with the respective variations of one standard deviation.

**Figure 8 sensors-20-05709-f008:**
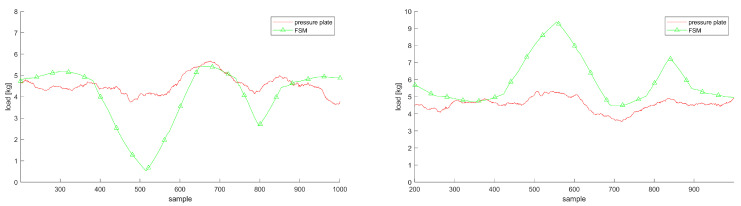
vGRF trajectories for the left and right foot demonstrating the effect of poorly tracked CoM while shifting the weight back and forth; both plots show the estimations and measurements in Newton per kilogram.

**Table 1 sensors-20-05709-t001:** Error metrics for the subregions, i.e., heel (${\mathrm{vGRF}}_{H}$), foot arch (${\mathrm{vGRF}}_{A}$), front foot (${\mathrm{vGRF}}_{F}$) and the total foot (vGRF), per movement for each foot. Walking with normal speed (NW), squatting (SQ), swaying all around (AR), from side to side (S2S) and back and forth (BF). The chosen metrics, i.e., root-mean-square error (RMSE), relative root-mean-square error (rRMSE), mean-absolute error (MAE), standard deviation (SD) and the correlation coefficient (*r*), are determined including all measured cycles per motion.

**AR**	**(Left Foot)**				**(Right Foot)**			
	${\mathrm{vGRF}}_{H}$	${\mathrm{vGRF}}_{A}$	${\mathrm{vGRF}}_{F}$	vGRF	${\mathrm{vGRF}}_{H}$	${\mathrm{vGRF}}_{A}$	${\mathrm{vGRF}}_{F}$	vGRF
RMSE [N/kg]	1.46	0.59	1.18	1.28	1.50	0.60	1.26	1.25
rRMSE [%]	18.04	20.63	14.83	13.28	20.39	20.53	17.04	13.15
MAE [N/kg]	1.03	0.39	0.89	1.01	1.03	0.41	0.92	1.03
SD [N/kg]	1.03	0.44	0.78	0.78	1.09	0.44	0.86	0.71
r	0.87	0.85	0.84	0.94	0.84	0.80	0.81	0.94
**S2S**	**(Left Foot)**				**(Right Foot)**			
	${\mathrm{vGRF}}_{H}$	${\mathrm{vGRF}}_{A}$	${\mathrm{vGRF}}_{F}$	vGRF	${\mathrm{vGRF}}_{H}$	${\mathrm{vGRF}}_{A}$	${\mathrm{vGRF}}_{F}$	vGRF
RMSE [N/kg]	2.00	0.64	1.48	1.31	2.36	0.76	1.64	1.41
rRMSE [%]	31.41	24.12	22.45	13.56	38.88	25.45	25.81	14.39
MAE [N/kg]	1.57	0.44	1.18	1.03	1.81	0.51	1.21	1.12
SD [N/kg]	1.23	0.46	0.90	0.81	1.52	0.57	1.10	0.86
r	0.76	0.85	0.66	0.93	0.78	0.80	0.68	0.93
**BF**	**(Left Foot)**				**(Right Foot)**			
	${\mathrm{vGRF}}_{H}$	${\mathrm{vGRF}}_{A}$	${\mathrm{vGRF}}_{F}$	vGRF	${\mathrm{vGRF}}_{H}$	${\mathrm{vGRF}}_{A}$	${\mathrm{vGRF}}_{F}$	vGRF
RMSE [N/kg]	1.33	0.34	1.23	0.75	1.30	0.49	1.18	0.60
rRMSE [%]	26.76	30.28	28.32	27.65	24.79	33.28	26.69	16.20
MAE [N/kg]	1.14	0.29	0.97	0.56	1.10	0.38	0.98	0.48
SD [N/kg]	0.68	0.19	0.76	0.50	0.69	0.31	0.67	0.36
r	0.91	0.47	0.92	0.65	0.86	0.17	0.92	0.75
**SQ**	**(Left Foot)**				**(Right Foot)**			
	${\mathrm{vGRF}}_{H}$	${\mathrm{vGRF}}_{A}$	${\mathrm{vGRF}}_{F}$	vGRF	${\mathrm{vGRF}}_{H}$	${\mathrm{vGRF}}_{A}$	${\mathrm{vGRF}}_{F}$	vGRF
RMSE [N/kg]	1.07	0.31	0.98	0.78	1.54	0.42	0.98	1.01
rRMSE [%]	22.93	20.07	27.54	14.96	32.14	20.97	16.74	18.82
MAE [N/kg]	0.87	0.21	0.77	0.61	1.31	0.31	0.82	0.77
SD [N/kg]	0.62	0.22	0.60	0.48	0.82	0.28	0.55	0.65
r	0.55	0.37	0.14	0.39	0.19	0.43	0.71	0.45
**NW**	**(Left Foot)**				**(Right Foot)**			
	${\mathrm{vGRF}}_{H}$	${\mathrm{vGRF}}_{A}$	${\mathrm{vGRF}}_{F}$	vGRF	${\mathrm{vGRF}}_{H}$	${\mathrm{vGRF}}_{A}$	${\mathrm{vGRF}}_{F}$	vGRF
RMSE [N/kg]	2.13	1.03	1.91	1.64	1.86	1.00	1.79	1.42
rRMSE [%]	24.77	33.19	20.67	15.94	22.71	31.42	19.40	13.60
MAE [N/kg]	1.51	0.74	1.55	1.22	1.36	0.75	1.42	1.09
SD [N/kg]	1.49	0.71	1.12	1.11	1.28	0.66	1.09	0.91
r	0.64	0.53	0.79	0.83	0.75	0.60	0.81	0.87

**Table 2 sensors-20-05709-t002:** Execution times regarding one full iteration (last line) and the substeps during a sway all around measurement. The third column shows the mean values and one standard deviation of the required time for 1308 executions.

Code Line	Procedure	Execution Time
5:	Determine $F_{\mathrm{GRF}}$	$2.1692\cdot 10^{-6}\pm 1.8414\cdot 10^{-5}\;s$
6–7:	Determine $\mu_{s,t},\Sigma_{s,t}$ and $f_{t}\left(x\right)$	0.047±0.0508s
9:	Register foot models	0.0087±0.0044s
10:	Determine weights ${\tilde{w}}_{t}\left(x\right)$	0.0545±0.002s
11–12:	Determine weights ${\hat{w}}_{\cdot ,t}$	$2.5707\cdot 10^{-4}\pm 1.4173\cdot 10^{-4}\;s$
14:	Integration and Normalization	$0.0036\pm 2.9285\cdot 10^{-4}\;s$
4–15:	One complete iteration	0.1141±0.0578s

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
