# Peer review of "Force Shadows: An Online Method to Estimate and Distribute Vertical Ground Reaction Forces from Kinematic Data"

_sensors, 2020, doi:10.3390/s20195709_

Round 1

Reviewer 1 Report

This paper presents the Force Shadow Method for distributing the vertical ground reaction force (GRF) computed only from kinematic motion data and mass distribution to multiple contact points. The main idea is to approximate the force distribution using projection of the COMs of individual links to the ground using the direction of the GRF. The method also utilizes heuristics from biomechanics such as foot shape and weight shifting for fine tuning. The method is compared to ground-truth pressure measurement with three subjects.

The paper contains some interesting insights on using biomechanical knowledge to supplement lack of data, and the technical content looks correct. The paper is also well structured and overall well written. I have a few comments:

  1. The results (especially Figs. 4 and 5) look fine, but without target accuracy it is impossible to know if these are good enough. Do you have a specific application in mind? Evaluation is even more difficult because y(k) is not defined (based on the unit I’m guessing weight-normalized load).
  2. One of the claims is that the method has low computational complexity, but there is no information about computation time.
  3. In addition to the GRF, you can also compute the reaction torque component to obtain the total center of pressure. Will this information help?
  4. I was a bit confused by the abstract because it states that the goal is “to reconstruct the vertical GRF” but computing the (total) GRF from motion data is not difficult. The main goal actually is to compute the distribution of the vertical GRF among multiple contact regions as correctly indicated in the title. I’d suggest editing the abstract accordingly.
  5. Line 29 “one contact point, e.g., during single support phase in walking”: there can be multiple contact points even in single support phase (as the authors show later by setting three contact regions per foot).
  6. The term “contact surface” sounds like the entire floor. I’d suggest using “contact region” instead.

Reviewer 2 Report

This paper introduces an online method to estimate and distribute vertical ground reaction forces from kinematic data. The authors proposed mathematical modeling approaches to estimate vertical ground reaction forces from kinematic data and evaluated the developed algorithm by comparing the estimated vertical ground reaction forces with the corresponding vertical ground reaction forces measured by a pressure plate. The authors chose various activities such as normal speed walking, squatting, swaying all around, swaying from side to side, and swaying back and forth. Several metrics (i.e., root-mean-square error (RMSE), relative RMSE, mean-absolute error, standard deviation, and correlation coefficient) were determined to quantify the similarity of the estimated vertical ground reaction forces with the measured one. Overall, the developed algorithm predicted reasonable vertical ground reaction forces in various activities. The topic of the present manuscript may be interesting for the readers in the fields. There are several issues to be addressed:

  1. Introduction: Why did the authors focus on the “vertical” ground reaction force? This reviewer agrees that the ground reaction force carries important information to analyze gait parameters in many applications. However, it is not clear how important the vertical ground reaction force itself would be. In general, vertical component would be important to increase the center of mass height, anteroposterior component to accelerate or break the center of mass thrust, and mediolateral component to maintain the center of mass balance. Thus, especially in clinical applications, anteroposterior or mediolateral component would be more of interest rather than vertical component. In the current introduction, the authors describe an importance of ground reaction forces, but focus on vertical ground reaction force only without solid rationale. Why is the estimation of vertical ground reaction force important? And what fields? Please clarify it.
  2. Line 81: How would the force shadow method be reasonably applied for clinical populations. For example, amputee or stroke. In many clinical populations, single stance time is likely different between both sides and, double support time on the affected side is likely shorter than on the non-affected side. Please address this point clearly.
  3. Line 84: What if the initial foot contact is with toe, rather than heel? Would the proposed contact surface estimation model work well as desired? This is because many clinical populations likely have a compensated walking pattern. Please discuss this point.
  4. Line 99: Again, how well would the hip flexion muscles model take into account a compensated walking pattern in clinical populations? Please address this point.
  5. Line 145: How much will an error in the vertical component of ground reaction forces result in an error in joint torques? Please include joint torque-related metrics.
  6. Line 183: Please address all issues above in the limitation or future work, if needed.
  7. This reviewer strongly recommends that the focus of this paper should be more specifically defined. Also, limitations of the proposed method should be clearly described.
  8. In general, there are several grammatical errors. Please proofread the manuscript carefully.
